# The MADS-Box Transcription Factor CaRIN Positively Regulates Chlorophyll Degradation During Pepper (*Capsicum annuum* L.) Fruit Ripening by Repressing the Expression of *CaLhcb-P4*

**DOI:** 10.3390/plants14030445

**Published:** 2025-02-03

**Authors:** Yingying Song, Qing Cheng, Xingzhe Li, Shijie Ma, Huolin Shen, Liang Sun

**Affiliations:** Beijing Key Laboratory of Growth and Developmental Regulation for Protected Vegetable Crops, Department of Vegetable Science, College of Horticulture, China Agricultural University, Beijing 100193, China

**Keywords:** *CaLhcb-P4*, carotenoid, chlorophyll, DAP-seq, fruit ripening, MADS-RIN, RNA-seq

## Abstract

Pepper (*Capsicum* spp.) is an important global vegetable and spice, with fruit color being a key determinant of its commercial quality. However, the regulatory mechanisms underlying pepper fruit color are still not fully understood. This study focuses on the MADS-RIPENING INHIBITOR (MADS-RIN), a MADS-box transcription factor that regulates various aspects of fruit ripening, including pigmentation. We identified *CaRIN*, a homolog of tomato’s *SlRIN*, whose expression is closely associated with fruit ripening in pepper. Silencing *CaRIN* through virus-induced gene silencing (VIGS) resulted in increased chlorophyll and chlorophyll a content, reduced carotenoid accumulation, and uneven fruit coloration. Integrative analysis of the RNA-seq and DAP-seq data identified 77 target genes regulated by CaRIN, which was involved in processes such as chlorophyll metabolism and plant hormone signaling. Yeast one-hybrid (Y1H) and dual-luciferase (LUC) assays demonstrated that CaRIN directly bound to the promoter of *CaLhcb-P4*, repressing its expression. Downregulation of *CaLhcb-P4* in pepper fruits via VIGS accelerated chlorophyll degradation. Additionally, CaRIN indirectly regulated multiple genes associated with chlorophyll and carotenoid metabolism, sugar transport, and cell wall degradation. These findings provide novel insights into the regulatory mechanisms of chlorophyll degradation during pepper fruit ripening, offering a foundation for further research and potential genetic improvement strategies.

## 1. Introduction

Pepper (*Capsicum* spp.) is a globally cultivated vegetable and spice, ranking among the top ten vegetables in terms of cultivated area in 2022. The quality of pepper fruits is determined by several traits that affect their appearance, flavor, texture, and nutritional value [1,2]. Among these, fruit color is especially important, as the pigments responsible for color are closely linked to both esthetic appeal and health benefits. Pepper fruits exhibit a diverse range of colors, which result from varying concentrations of carotenoids, chlorophylls, anthocyanins, and other pigments [3]. Carotenoids are the primary contributors to the color of ripe pepper fruits, while chlorophylls and anthocyanins play a more prominent role in immature fruits, though they can also influence the color of ripe fruits [4]. Therefore, understanding the molecular regulatory mechanisms underlying the accumulation of these pigments, particularly chlorophylls and carotenoids, is crucial for genetic improvement and enhancing fruit quality in pepper.

The ripening process in most fruits involves the conversion of chloroplasts into chromoplasts, accompanied by a distinct color change. This transition occurs through three main mechanisms: (I) the degradation of chlorophyll, which is linked to the breakdown and recycling of thylakoid membranes and photosynthetic proteins; (II) the accumulation of colored carotenoids in developing chromoplasts or other plastid-derived structures; and (III) the accumulation of anthocyanins or flavonoid-like pigments in cytosolic vesicles [5,6,7,8,9,10]. In tomato, chloroplasts dominate during the early stages, giving the fruit a green appearance. As the fruit matures, chlorophyll and thylakoid membranes degrade, revealing carotenoids and anthocyanins, which contribute to the diverse colors of ripe fruit, such as red, yellow, orange, and purple [11,12]. In citrus, chlorophyll degradation is accompanied by an increase in carotenoids, while in Golden Delicious apples, the degradation of chlorophyll results in a yellow color [13]. Similarly, in pepper, the transition from green, purple, or black immature fruits to yellow, orange, or red ripe fruits is driven by chlorophyll and anthocyanin degradation as well as carotenoid accumulation [14,15].

Recent studies have identified numerous transcription factors (TFs) that regulate pigment accumulation in fruits. In tomato, MADS-box proteins, particularly the MADS-RIPENING INHIBITOR (MADS-RIN) gene, play key roles in the transcriptional regulation of fruit ripening, including chlorophyll degradation and carotenoid accumulation [16]. The *rin* mutation in tomato results in a chimeric gene formed by the fusion of the *RIN* and *MACROCALYX* (*MC*) genes, producing a non-functional RIN-MC protein that inhibits ripening and carotenoid accumulation [17]. However, CRISPR/Cas9-mediated knockout of *RIN* did not fully suppress ripening initiation, indicating complex genetic regulation of ripening [18]. In addition to MADS-box proteins, other TFs such as MYB, WRKY, LOL, and BBX have also been identified as regulators of chlorophyll and carotenoid content [19,20,21]. These findings suggest that TFs act synergistically to regulate both chlorophyll degradation and carotenoid biosynthesis. However, the genetic network governing color transitions during pepper fruit ripening remains largely unexplored.

## 2. Results

### 2.1. Silencing of CaRIN Inhibits Chlorophyll Degradation and Carotenoid Accumulation During Pepper Fruit Ripening

BLAST analysis was conducted to identify homologs of *SlRIN*, resulting in the identification of five *MADS-box* genes. Phylogenetic analysis revealed that CaMADS3 and CaRIN share the closest evolutionary relationship with SlRIN (Figure 1A). qRT-PCR analysis showed that CaRIN exhibited higher expression levels, and its expression pattern closely correlated with the progression of fruit ripening (Figure 1B), suggesting a role for *CaRIN* in pepper fruit development. To investigate the function of *CaRIN* in fruit ripening, its expression was downregulated using VIGS (Figure 1C). The TRV2 vector used in this study harbors a GFP expression cassette, allowing for the selection of seedlings that fluoresced under UV light two weeks after vacuum infiltration (Figure 1D). Positive control fruits showed significant photobleaching at the mature green stage, confirming the effectiveness of the TRV-based VIGS system for studying gene function in pepper fruit development (Figure 1D). qRT-PCR analysis demonstrated that *CaRIN* transcript levels in silenced fruits were reduced to ~65% of those in the negative control (pTRV2-GFP) (Figure 1E). Silenced fruits also displayed uneven pigmentation at 64 DPA, with some areas of the fruit peel remaining brown, while the negative control fruits exhibited uniform red coloration at the same developmental stage (Figure 1E). In line with these color changes, total chlorophyll and chlorophyll a content in the pericarp of *CaRIN*-silenced fruits were significantly higher (by over 45%) compared to the negative control (Figure 1E). However, chlorophyll b content did not differ significantly between the two groups (Figure 1E). In contrast, carotenoid contents in the pericarp of *CaRIN*-silenced fruits were reduced by ~40% at 64 DPA compared to the negative control (Figure 1E). These results indicate that *CaRIN* is involved in regulating both chlorophyll degradation and carotenoid biosynthesis during pepper fruit ripening.

### 2.2. Impact of CaRIN on Transcriptome of Pepper Fruit During Ripening

To identify genes transcriptionally regulated by *CaRIN* during pepper fruit ripening, a comparative transcriptome analysis was conducted using fruits collected from *CaRIN*-silenced and negative control plants at 64 DPA. Principal component analysis (PCA) of the gene expression data clearly distinguished samples collected from the silenced and control fruits, indicating a significant impact of *CaRIN* on the fruit transcriptome (Figure 2A). Using DESeq2, a total of 2960 differentially expressed genes (DEGs) were identified, with 1364 genes downregulated and 1596 genes upregulated in the *CaRIN*-silenced fruits (Figure 2B; Appendix A). Gene ontology (GO) enrichment analysis revealed three distinct ontologies that represent key aspects of protein function and the most dominant biological processes, including “molecular function”, “response to abscisic acid”, “response to salt stress”, “chloroplast organization”, “chloroplast stroma”, and “carotenoid metabolism” (Figure 2C).

To further investigate the regulatory effects of *CaRIN* on chlorophyll metabolism and carotenoid accumulation, genes associated with these processes were screened out. For chlorophyll metabolism-related genes, the transcript levels of *CaNYC1* (encoding chlorophyll(ide) b reductase), *CaSGR* (encoding STAY-GREEN), *CaUROS* (encoding uroporphyrinogen III synthase), and *CaCHLD* (encoding magnesium-chelatase subunit ChlD) were significantly reduced in the *CaRIN*-silenced fruits compared to the negative control. In contrast, the transcript levels of *CaCHLI* (encoding magnesium-chelatase subunit ChlI), *CaLhcb7* (encoding chlorophyll a-b binding protein 7), *CaPOR1* (encoding protochlorophyllide reductase), *CaLhcb-P4* (encoding chlorophyll a-b binding protein, part of the light-harvesting complex), *CaCPO* (encoding oxygen-independent coproporphyrinogen-III oxidase-like protein), and *CaUROD* (encoding putative uroporphyrinogen decarboxylase) were elevated in the silenced fruits (Figure 2D). For carotenoid biosynthesis-related genes, the expression of *CaZEP* (encoding zeaxanthin epoxidase) and *CaLcyB* (encoding lycopene beta cyclase) was upregulated in the silenced fruits, while the transcript levels of *CaCCS* (encoding capsanthin/capsorubin synthase), *CaCrtZ1* (encoding beta-carotene hydroxylase 1), and *CaPSY1* (encoding 15-cis-phytoene synthase) were downregulated (Figure 2E).

Thus, the abovementioned results suggest that *CaRIN* plays a crucial role in the transcriptional regulation of pepper fruit ripening, particularly in chlorophyll and carotenoid metabolism.

### 2.3. Identification of CaRIN Binding Sites in Pepper Using DAP-Seq

To further investigate *CaRIN*-mediated transcriptional regulation during pepper fruit ripening, DAP-seq was performed to identify the CaRIN binding sites at the genomic level. From two technical replicates, 12,398 enriched peaks corresponding to 868 genes were identified as high-confidence CaRIN binding sites, which were predominantly located upstream of the transcription start site (TSS) (Figure 3A,B; Appendix A). These binding sites were distributed relatively uniformly across all 12 chromosomes of the pepper genome (Figure 3C). An analysis of cis-regulatory regions revealed that CaRIN binding occurred in promoters (3.05%), 5′ UTRs (0.18%), 3′ UTRs (0.06%), exons (0.31%), introns (3.79%), and intergenic regions (92.48%) (Figure 3D).

In order to gain deeper insights into the DNA-binding characteristics of CaRIN, its target motifs were predicted using DAP-seq data. Among the identified motifs, the canonical CC(A/T)_6_GG recognition sequence stood out, which is a well-established binding site for RIN, a key member of the MADS-box TF family (Figure 3E). Centrality analysis showed that this motif was distributed in the binding peaks detected by DAP-seq, following a normal distribution pattern (Figure 3E). GO enrichment analysis of the target genes identified the top enriched terms, including “translation”, “chloroplast thylakoid membrane”, “response to cold”, and “chloroplast” (Figure 3F). Notably, several ripening-related regulators were identified among the putative target genes, including two cell wall-associated genes, *CaCSLD2* (encoding cellulose synthase-like protein D2) and *CaCesA3* (encoding cellulose synthase A catalytic subunit 3); a sugar transport gene, *CaSTP14* (encoding sugar transport protein 14-like); and several chlorophyll-related genes, such as *CaCRS1* (encoding chloroplastic group IIA intron splicing facilitator), *CaUROS* (encoding uroporphyrinogen-III synthase), *CaPSAB* (encoding photosystem I P700 chlorophyll A apoprotein A2), and *CaLhcb-P4* (encoding chlorophyll a-b binding protein P4). Additionally, several transcription factors (TFs) were identified, including WRKY TFs (CaWRKY49, CaWRKY55, CaWRKY18, CaWRKY17, CaWRKY75), MADS-box TFs (CaAGL62-like, CaAGL62, CaAGL29, CaSOC1), and NAC66-like (Appendix A).

### 2.4. Identification of CaRIN-Regulated Target Genes Through Integrated Analysis of RNA-Seq and DAP-Seq

DAP-seq identified 944 genes with CaRIN binding peaks within 2 kb of the TSS, while RNA-seq detected 3037 DEGs. Among these, 77 genes were common to both datasets, representing strong candidate targets of CaRIN (Figure 4A; Appendix A). Of these, 50 genes were significantly downregulated and 27 upregulated in the silenced fruits, suggesting that CaRIN positively regulates the former and negatively regulates the latter (Figure 4A).

KEGG pathway enrichment analysis revealed that the 77 candidates were involved in several metabolic pathways, including “plant hormone signal transduction”, “fatty acid metabolism”, “porphyrin and chlorophyll metabolism”, and “starch and sucrose metabolism” (Figure 4B). Consistent with the phenotype of the *CaRIN*-silenced fruits, two key chlorophyll-related genes were identified: *CaUROS*, which encodes uroporphyrinogen-III synthase involved in chlorophyll biosynthesis, and *CaLhcb-P4*, which encodes a chlorophyll a/b binding protein crucial for light absorption, energy transfer to PSI and PSII, and thylakoid membrane maintenance. Additionally, a cell wall-related gene, *CaPEL13*, encoding pectate lyase 13, and a sugar metabolism-related gene, *CaSTP14*, encoding sugar transport protein 14, were also identified, indicating that CaRIN may influence cell wall pectin content and sugar transport in pepper fruit during ripening. Apart from the abovementioned metabolic genes, several TFs, including CaWRKY75, were also identified, suggesting that CaRIN may regulate gene expression indirectly.

**Figure 3 plants-14-00445-f003:**
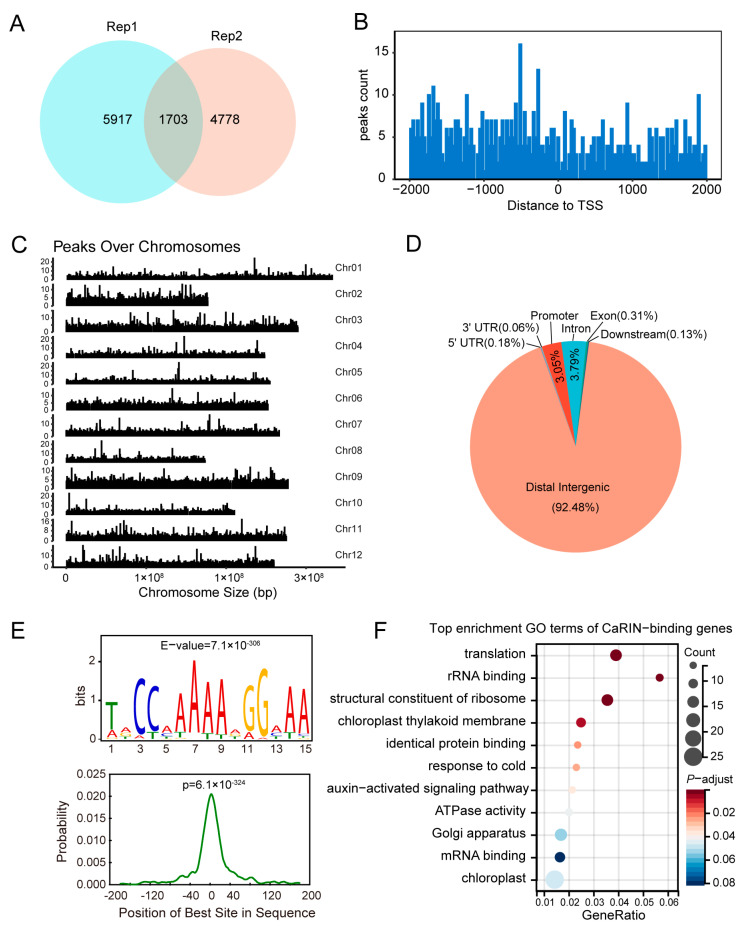
The genome-wide identification of CaRIN binding sites via DAP-seq. (**A**) A Venn diagram of the two biological replicates of DAP-seq. (**B**) The distribution of CaRIN binding sites relative to the transcription start sites (TSS). (**C**) The distribution of CaRIN binding sites across the 12 chromosomes. (**D**) The localization of CaRIN binding sites in relation to cis-regulatory regions. (**E**) The primary motif bound by CaRIN. (**F**) The top enriched GO terms associated with CaRIN-bound genes.

**Figure 4 plants-14-00445-f004:**
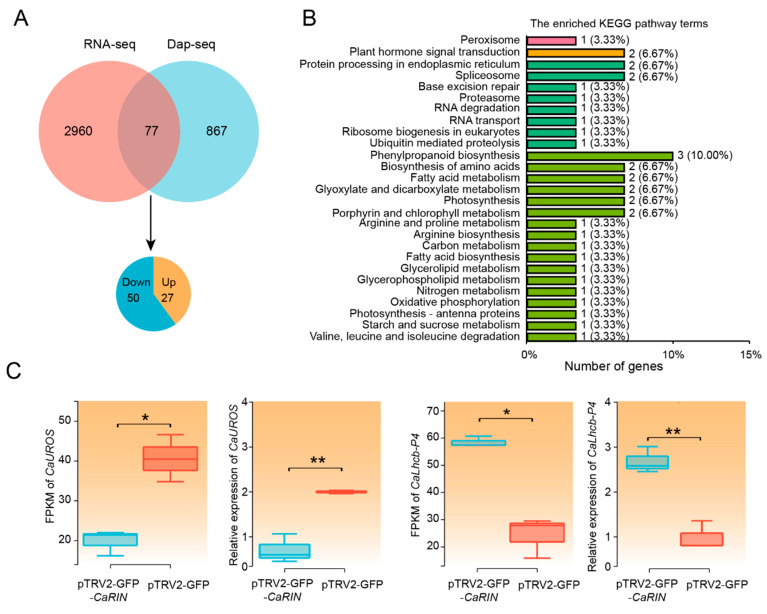
The Identification of *CaRIN*-regulated target genes. (**A**) A Venn diagram showing the overlap between *CaRIN*-bound genes identified by DAP-seq and DEGs from RNA-seq. (**B**) The top KEGG pathways enriched in the *CaRIN*-regulated target genes. (**C**) The relative expression levels of *CaUROS* and *CaLhcb-P4* in *CaRIN*-silenced and negative control fruits (* *p* < 0.05; ** *p* < 0.01).

### 2.5. CaRIN Directly Regulates Key Genes in Chlorophyll Metabolism Pathway

In order to validate the direct regulation of chlorophyll metabolism by CaRIN, Y1H assays were performed to assess the binding of CaRIN to the promoters of *CaLhcb-P4* and *CaUROS*, which were identified in the integrated analysis of RNA-seq and DAP-seq (Figure 5A). The results revealed that yeast cells transformed with pJG4-5-*CaRIN* and pLacZi-pro*CaLhcb-P4* formed blue colonies on selective medium, while those transformed with pJG4-5-*CaRIN* and pLacZi-pro*CaUROS*, as well as the negative control, did not (Figure 5B), demonstrating that CaRIN specifically binds to the *CaLhcb-P4* promoter in yeast. Subsequently, a LUC reporter assay was employed to investigate the effect of CaRIN on the transcriptional regulation of *CaLhcb-P4*. The CDS of *CaRIN* was cloned into the pGreenII 62-SK vector (effector vector), while a 502 bp fragment of the *CaLhcb-P4* promoter region, including the CrAG-BOX element, was cloned into the pGreenII 0800-LUC vector (reporter vector) to generate the LUC reporter plasmid (Figure 5C). Co-infiltration of Agrobacterium strains containing 35S::*CaRIN* and *proCaLhcb*-*P4*::LUC led to a significant reduction in fluorescence intensity compared to the control, where the empty vector and *proCaLhcb*-*P4*::LUC were co-infiltrated (Figure 5D). The LUC/REN ratio in leaves co-expressing *CaRIN* and *proCaLhcb-P4*-LUC was notably lower than that in the control leaves. Furthermore, silencing *CaLhcb-P4* via VIGS using the injection method resulted in lightening of the green color of the pericarp (Figure 5F). The expression level of *CaLhcb-P4* in the silenced fruits was significantly reduced, and the chlorophyll a content was also notably lower, while the chlorophyll b content showed no significant difference. These findings suggest that the downregulation of *CaLhcb-P4* reduces chlorophyll content, primarily through the reduction in chlorophyll a levels.

Taken together, the abovementioned findings indicate that CaRIN plays a critical role in chlorophyll metabolism by directly repressing the expression of *CaLhcb-P4* in pepper fruit during ripening (Figure 6).

## 3. Discussion

MADS-RIN (or RIN) has been shown to be an important transcription factor in the regulation of fruit ripening in various horticultural crops, with one of its primary functions being the control of fruit coloring. In this study, based on sequence similarity and gene expression levels and patterns (Figure 1A,B), CaRIN is considered to be the most homologous gene to SlRIN in terms of function. Previous studies have also shown that overexpression of this gene in the tomato *rin* mutant partially rescued the phenotype of the mutant that is unable to ripen, indicating that *CaRIN* has the basic function of *SlRIN* [22]. In this experiment, the expression of *CaRIN* was downregulated using VIGS, and the most significant phenotype, which was visibly detectable, was the prolonged fruit color change process and uneven coloring (Figure 1D). The direct reason for this was the inhibition of chlorophyll degradation in fruit (Figure 1E). Moreover, although there was no significant difference in visual observation, the accumulation of carotenoids in the silenced fruits was also suppressed (Figure 1E). This phenomenon is similar to results obtained in tomato, apple, peach, and banana. In these horticultural crops, either natural mutations or artificial silencing of *SlRIN* and its homologous genes inhibits the degradation of chlorophyll and the accumulation of carotenoids or anthocyanins during ripening [11,23,24,25]. Building on the results discussed above, it can be inferred that the *RIN*-mediated regulation of color change during fruit ripening is highly conserved across both climacteric fruits (such as tomato, apple, peach, and banana) and non-climacteric fruits (such as pepper), as well as in dicot fruits (tomato, apple, peach, and pepper) and monocot fruits (banana). Among the various processes involved in color regulation during ripening, the most conserved mechanism appears to be the degradation of chlorophyll. This is because the primary pigments in the ripening fruits of tomato, pepper, and the peel of banana are carotenoids, while the key pigments in the ripening fruits of apple and peach are predominantly anthocyanins. Nonetheless, chlorophyll degradation is a common feature in all these fruits.

To further reveal the regulatory mechanism of *RIN* homologous genes in chlorophyll metabolism in non-climacteric pepper fruits, this study employed a series of omics, interaction, and regulatory experiments and found that CaRIN can directly bind to the promoter of the *CaLhcb-P4* gene and inhibit its expression. *CaLhcb-P4* encodes a chlorophyll a/b binding protein involved in light capture, transferring energy to the PS I and PS II centers, regulating the distribution of excitation energy, and maintaining thylakoid membrane structure. In this study, silencing this gene via VIGS reduced chlorophyll content in the fruit, ultimately leading to a delay in fruit de-greening. Similar results have also been observed in many studies; for example, downregulation of *Lhcb* homologous genes in *citrus* peel, cotton seedlings, and *Arabidopsis* leaves has been shown to significantly decrease chlorophyll content [26,27,28]. Therefore, it can be presumed that the upregulation of *CaLhcb-P4* due to the silencing of *CaRIN* in this study is one of the main reasons for the delayed chlorophyll degradation in the *CaRIN*-silenced fruits. Furthermore, it is noteworthy that in the *CaRIN*-silenced fruits, many chlorophyll metabolism-related genes also exhibited significant changes in expression, even though these genes were not identified as the direct targets of CaRIN in DAP-seq. For example, the expression levels of *CaNYC1*, *CaSGR*, *CaUROS*, and *CaCHLD* were significantly downregulated, while the expression levels of *CaCHLI*, *CalhcB7*, *CaPOR1*, *CaCPO*, and *CaUROD* were significantly upregulated, in the *CaRIN*-silenced fruits. This suggests that CaRIN may indirectly regulate the expression of these genes. *UROS*, *UROD*, *CHLD*, *CHLI*, *CHLM*, and *POR* are involved in chlorophyll synthesis, while *NYC* catalyzes the conversion of chlorophyll b into free chlorophyll a, thereby maintaining the stability of the photosystem [29]. Studies have shown that mutations in the *NYC* gene inhibit chlorophyll degradation, leading to increased chlorophyll accumulation and the formation of darker green fruits [29]. Additionally, *SGR* is another important gene in the chlorophyll pathway in green plants. Mutations in the *SGR* genes in *citrus*, pak choi, and Arabidopsis inhibit chlorophyll degradation [30,31,32,33]. Therefore, the expression changes in chlorophyll metabolism-related genes induced indirectly by *CaRIN* are another major reason for the delayed chlorophyll degradation and uneven coloring in the *CaRIN*-silenced fruits.

**Figure 6 plants-14-00445-f006:**
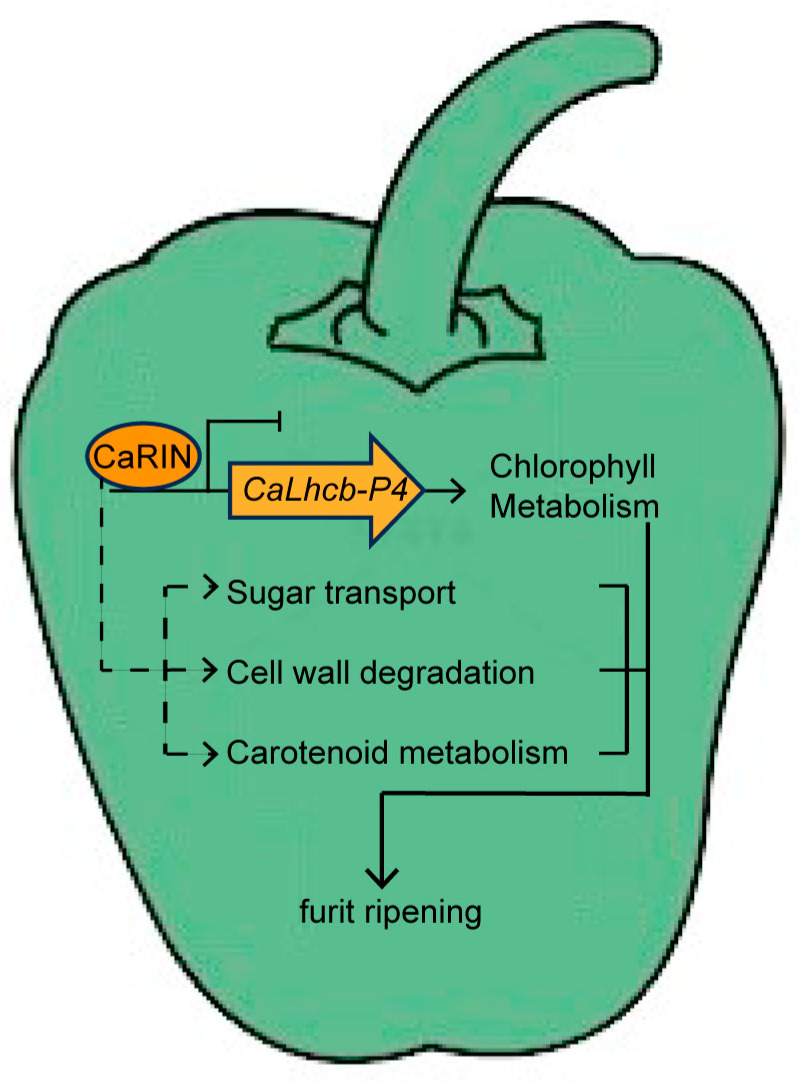
A proposed model for the regulation of fruit ripening by CaRIN in pepper. CaRIN represses *CaLhcb-P4* expression by binding the *CaLhcb-P4* promoter, and then, *CaLhcb-P4* influences chlorophyll metabolism. CaRIN also indirectly regulates multiple genes associated with chlorophyll and carotenoid metabolism, sugar transport, and cell wall degradation.

In addition to chlorophyll, the carotenoid content in the silenced fruits was significantly reduced, primarily due to the decreased expression of three key genes involved in carotenoid biosynthesis, including *CaPSY1*, *CaCCS*, and *CaCrtZ1*. Previous studies have shown that mutations in any of these genes can cause a color shift in the fruit from red to orange [1,34]. In this study, although DAP-seq did not identify the binding sites for carotenoid-related genes, it suggests that CaRIN may indirectly regulate the expression of genes related to carotenoid synthesis during pepper fruit development, thereby influencing carotenoid synthesis.

Besides pigment-related genes, hundreds of direct target genes of RIN have been identified in other horticultural crops, including ethylene biosynthesis genes such as *ACS2* and *ACS4* [35]; the ethylene receptor gene *NR/ETR3* [36]; cell wall-softening genes such as *PG*, *TBG4*, *EXP1*, and *CEL2* [35]; oxylipin flavor pathway genes such as *LoxC* and *ADH2* [37]; ubiquitin-proteasome degradation genes such as *SlUBC32* and *PSMD2* [38]; and sugar metabolism genes like *TAI* and *LIN5* [39,40]. In this study, through a combined analysis of DAP-seq and RNA-seq, 77 target genes were identified, including cell wall-related genes such as *CaPEL13* (encoding pectate lyase), *CaCel1* (encoding endoglucanase), *CaEXLA10* (encoding expansin-A10 precursor), 2 *CaEXLAs* (encoding expansin), and 2 CaPGs (encoding polygalacturonase); sugar metabolism-related genes such as *CaSTP14* (encoding sugar transport protein) and *CaYfbT* (encoding sugar phosphatase); and an ethylene signaling gene, *CaETR2* (encoding ethylene receptor 2) (Appendix A). However, genes related to ethylene synthesis, the oxylipin flavor pathway, and ubiquitin-proteasome degradation were not identified. These findings suggest that, in both climacteric and non-climacteric fruits, the regulation of the cell wall, sugar metabolism, and chloroplast metabolism by CaRIN is highly likely to be relatively conserved.

In conclusion, by silencing *CaRIN* in pepper, we identified a novel direct target gene, *CaLhcb-P4*, which had not been previously reported. CaRIN directly binds to the promoter of *CaLhcb-P4* and inhibits its expression, thereby affecting the changes in chlorophyll content during fruit ripening and suppressing the conversion of chloroplasts into chromoplasts. Additionally, through DAP-seq, we identified other target genes enriched in various biological processes, confirming the diversity and complexity of the CaRIN regulatory network. This study provides new insights into the complex transcriptional regulatory mechanisms of *CaRIN* in pepper fruit development and offers potential strategies for molecular breeding in the pepper industry.

## 4. Materials and Methods

### 4.1. Sequence Alignment and Phylogenetic Tree Construction

The TFs with the highest sequence identity to *CaRIN* and other known MADS TFs were retrieved from the public databases NCBI (https://www.ncbi.nlm.nih.gov, accessed on 12 November 2024) and the Pepper Genomics Database (http://ted.bti.cornell.edu/cgi-bin/pepper/index, accessed on 12 November 2024) to construct a phylogenetic tree. Sequence alignment and phylogenetic tree construction were performed using clustalx 2 (https://www.ebi.ac.uk/Tools/msa/clustalw2/, accessed on 12 November 2024) with 1000 bootstrap trials.

### 4.2. Virus-Induced Gene Silencing

A Tobacco Rattle Virus (TRV) system containing the GFP (Green Fluorescent Protein) reporter gene was used for functional studies [41]. The online tool Sol Genomics Network (https://vigs.solgenomics.net/, accessed on 15 November 2024) was used for partial sequence selection. Target fragments of *CaRIN* (Caz04g14950) and *CaPDS* (the phytoene desaturase gene that served as a positive control) were amplified using specific primers and cloned into the pTRV2-GFP vector. The VIGS experiments were conducted following the methods outlined in a previous study [42]. In brief, pepper seedlings were infiltrated with Agrobacterium tumefaciens GV3101 strains carrying both pTRV2-GFP-*CaRIN* and pTRV1 in a ratio of 1:1 (v/v), or with pTRV1 and pTRV2-GFP (the negative control). Photobleaching was observed in the positive control, and UV was used to observe pTRV spread and monitor VIGS efficiency about 2 weeks after infiltration. Total RNA was extracted from the peel of red ripe fruits for gene expression analysis and chlorophyll determination. The primers used for vector construction are listed in Appendix A.

### 4.3. Transcriptome Analysis

Total RNA was extracted using Trizol from three biological replicates of fruits at the red ripe stage for gene expression analysis. As above, RNA-seq libraries were constructed and sequenced on the NovaSeq platform. The clean reads were mapped against the GS genome using Hisat2 (v2.0.5) software. The number of reads mapped was counted using HTSeq (v0.6.1), and the fragments per kilobase of exon per million mapped (FPKM) fragments values were subsequently calculated for each gene. Differentially expressed genes (DEGs) were identified in pTRV2-GFP-*CaRIN* and pTRV2-GFP using the DESeq R package, with the threshold for significant differential expression set to a fold change ≥1.5 and the *p*-value set to ≤0.05.

### 4.4. Total Chlorophyll and Carotenoid Measurement

To determine the chlorophyll and carotenoid contents, approximately 1.0 g of pericarp samples were placed into tubes containing 95% ethanol and then frozen. The frozen pericarp samples were placed in a mortar and ground with anhydrous ethanol until only the white tissue remained. The liquid from the macerated tissue was filtered through filter disks. The absorbance (A) was measured using a spectrophotometer at wavelengths of 470 nm, 649 nm, and 665 nm, and the calculation equations followed those described previously [43].

### 4.5. DAP-Seq

In vitro DAP-seq was performed as described previously [44]. Genomic DNA was extracted from ripe pericarps, and an affinity purification library was constructed from fragmented DNA using the NGS0602-MICH TLX DNA-Seq Kit following the manufacturer’s recommendations. For cell-free protein expression in vitro, Halo-tagged TF CaRIN and recombinant protein were produced in a TNT SP6 Coupled Wheat Germ Extract System (Promega, Madison, WI, USA). We incubated the cell-free protein expression premix and protein expression vector together for 2 h at 25 °C according to the manufacturer’s recommendations. We then detected specific bands of target protein after determining the protein expression. The Halo-tagged TF was bound to Beads and incubated with the affinity purification library. The typical motifs in the peak regions were analyzed using the MEME-chip software (v5.3.3) [45]. Peak-related genes were annotated using the R package ChIPseeker [46]. The frequency of distribution of reads in the vicinity of the TSS was analyzed using depTools2 software (v3.3.0), and density profiles and corresponding heatmaps were plotted [47]. The frequency of the distribution of peaks around TSS was calculated to visualize the distribution of peaks across the chromosome.

### 4.6. Y1H Assay

The promoter fragments of *CaLhcb-P4* (496 bp) and *CaUROS* (447 bp) containing CaRIN binding peaks (up to 2 kb upstream from a TSS) were cloned into the pLacZi target vector to construct the pLacZi-*CaLhcb-P4* and pLacZi-*CaUROS* recombinant vectors. The CDS of *CaRIN* was cloned into the pB42AD vector to construct the pB42AD:*CaRIN* expression vector. Both vectors were co-transformed into EGY48 yeast cells. The yeast was cultured in SD-Trp-Ura double-deficiency medium for 2 to 3 days. Single clones were transferred to SD-Trp-Ura medium containing X-Gal and incubated at 30 °C for 2 to 3 days to observe the color change in yeast cells [48]. The primers used for vector construction are listed in Appendix A.

### 4.7. Dual-Luciferase Reporter Assay

An LUC reporter gene transient detection assay was performed in tobacco (*N. benthamiana*). The CDS of *CaRIN* was inserted into the pGreenII 62-SK vector (effector vector) driven by the 35S promoter. The same promoter fragment of *CaLhcb-P4* (496 bp) as in YIH was inserted into the pGreenII 0800-LUC vector (reporter vector). The recombinant plasmids were transformed into *A. tumefaciens* GV3101 (P19). Agrobacterium was cultured to an OD600 value of 0.8–1.0 and suspended in osmotic buffer (10 mM MES, 10 mM MgCl_2_, 150 μM acetylcoumarin, pH = 5.8). The effector vector and reporter vector were mixed together at an OD600 ratio of 6:4 and injected into abaxial surfaces in tobacco leaves. Luciferase activity images of the leaves were observed 48 to 72 h post-transformation using the Night SHADE LB 985 system (Berthold, Bad Wildbad, Germany), and the wavelength was set to 350 to 700 nm. The luciferase activity was quantified using an LUC reporter assay kit as described previously [49], and the transcriptional activation ability of *CaRIN* was determined according to the LUC/REN ratio based on three biological replicates. The primers used for vector construction are listed in Appendix A.

### 4.8. qRT-PCR Analysis

Total RNA was extracted from pericarp samples using TRIzol (Thermo Fisher Scientific, Waltham, MA, USA) reagent. For reverse transcription PCR, the first-strand cDNA was synthesized from 0.8 μg of total RNA using a HiScript IV 1st Strand cDNA Synthesis Kit (Nanjing Vazyme Biotech Co., Ltd., Nanjing, China). qRT-PCR was performed using the SupRealQ Purple Universal SYBR qPCR Master Mix (U+) (Nanjing Vazyme Biotech Co., Ltd., Nanjing, China). *CaUBI-3* was used as the housekeeping gene, and the relative expression values were calculated using the 2^–∆∆CT^ method. For each gene, each biological replicate was set with three independent technical replicates. The primer sequences used are listed in Appendix A.

### 4.9. Statistical Analysis

For all experiments, three independent repetitions were conducted. Statistical analyses were performed using SPSS (IBM SPSS Statistics, v.22; SPSS Inc., Chicago, IL, USA) software and Microsoft EXCEL 2010. Significant differences were calculated using one-way ANOVA, Tukey’s test, and Student’s *t*-test (* *p* < 0.05; ** *p* < 0.01; *** *p* < 0.001).

## Figures and Tables

**Figure 1 plants-14-00445-f001:**
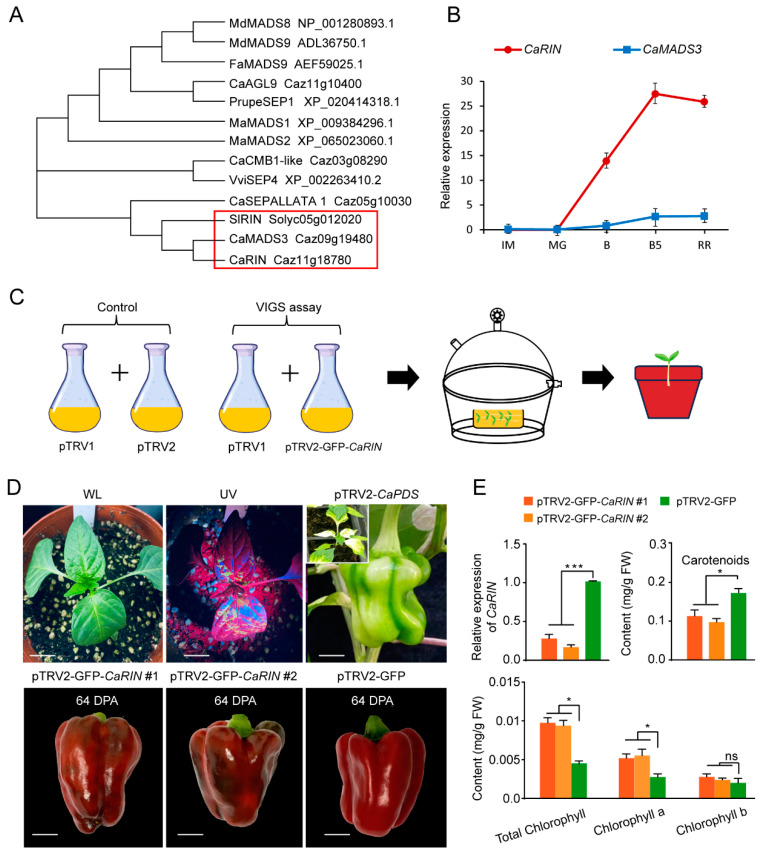
*CaRIN* regulates chlorophyll and carotenoid contents in pepper fruit during ripening. (**A**) A phylogenetic tree of SlRIN and its homologs in other species. (**B**) The relative expression levels of *CaRIN* and *CaMADS3* in the sweet pepper inbred line 16C391 at various developmental stages: IM (immature green, 22 DPA), MG (mature green, 40 DPA), B (breaker, 49 DPA), B5 (breaker + 5, 54 DPA), and RR (red ripe, 63 DPA). (**C**) A schematic diagram of VIGS using the vacuum infiltration method. pTVR1 + pTRV2-GFP serves as the negative control, while pTVR1 + pTRV2-CaPDS is used as the positive control. (**D**) Phenotypic observations of *CaRIN*-silenced seedlings under white light and UV (top left and middle), light bleaching in the *CaPDS*-silenced fruit at the mature green stage (top right), *CaRIN*-silenced fruits at the red ripe stage (bottom right and middle), and fruit from the negative control (bottom left). Scale bar = 1.5 cm. (**E**) The relative expression of *CaRIN* and levels of chlorophyll and carotenoids in fruits at 64 DPA from the *CaRIN*-silenced and negative control plants. The data represent the mean ± SE (n = 3). Significant differences were determined using Student’s *t*-test (* *p* < 0.05; *** *p* < 0.001; ns, not significant).

**Figure 2 plants-14-00445-f002:**
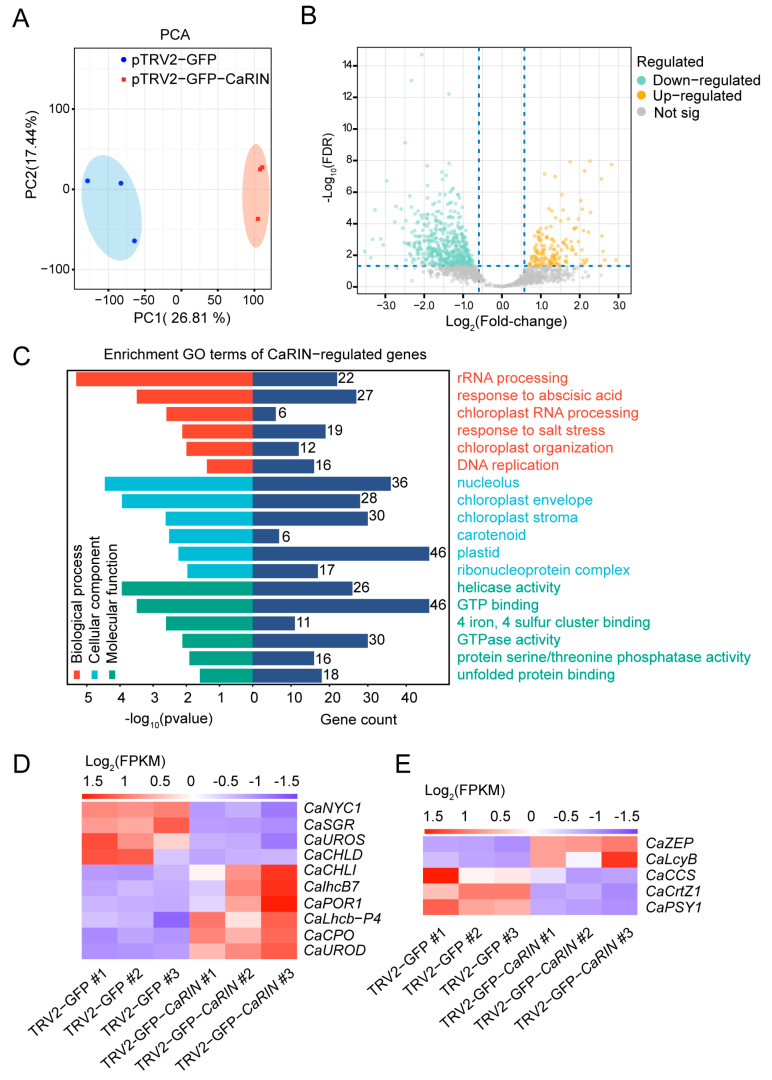
RNA-seq analysis of the pericarp collected from *CaRIN*-silenced and negative control fruits. (**A**) PCA of RNA-seq data. (**B**) Volcano plots of the DEGs. (**C**) GO enrichment analysis of DEGs. (**D**,**E**) heatmaps of DEGs related to chloroplast metabolism and carotenoid synthesis. #1, #2 and #3 represent three biological replicates.

**Figure 5 plants-14-00445-f005:**
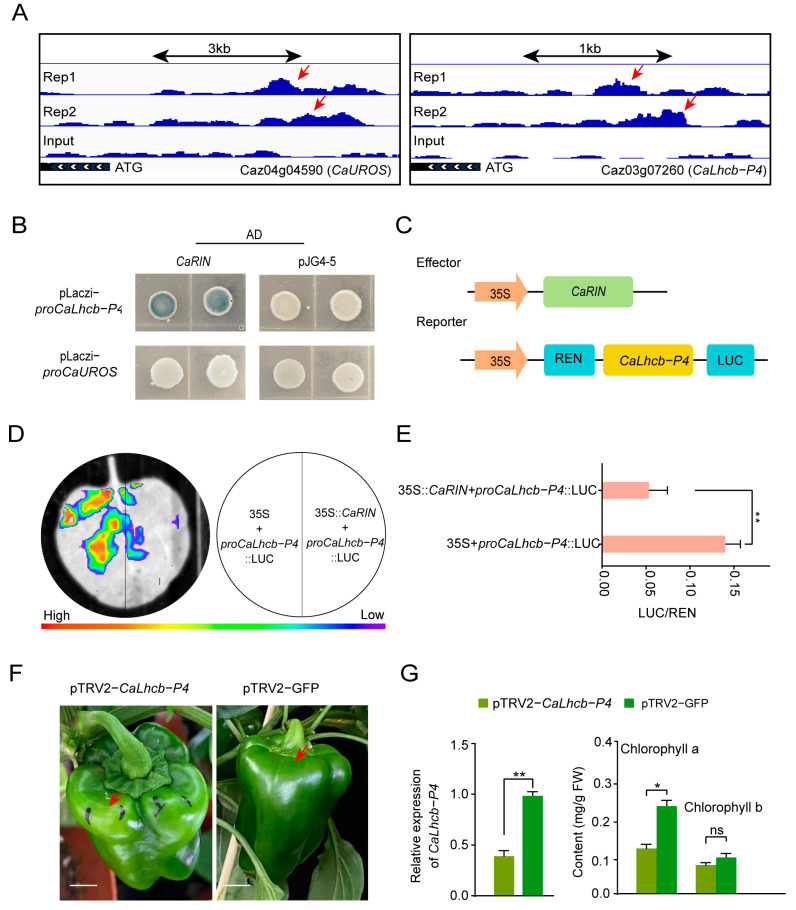
The interaction of CaRIN with *CaLhcb-P4* and the silencing of CaLhcb-P4 in pepper fruit. (**A**) The binding peaks of CaRIN (Repeats 1 and 2) and the negative control (input) at the promoter regions of *CaLhcb-P4* and *CaUROS*, as determined by DAP-seq. Red arrows represent binding peaks. (**B**) The Y1H assay of CaRIN interaction with the promoters of *CaLhcb-P4* and *CaUROS*. Three independent yeast colonies were tested for each treatment, and the results were observed 3 days post-treatment. (**C**) A schematic representation of the reporter and effector constructs used in the dual-luciferase assay. (**D**) The LUC imaging assay confirming that CaRIN directly represses the expression of *CaLhcb-P4*. (**E**) The LUC/REN ratio analysis for the detection of CaRIN targeting *CaLhcb-P4*. LUC, firefly luciferase activity; REN, Renilla luciferase activity. The data are presented as mean values ± SD from three independent biological replicates. (**F**) The typical phenotype of CaLhcb-P4-silenced and control fruits. Red arrows represent silencing position. Scale bar = 1.5 cm. (**G**) The relative expression of *CaLhcb-P4* and chlorophyll content in *CaLhcb-P4*-silenced and the negative control fruits. The data are expressed as the mean ± SE (n = 3). Statistical significance was assessed by Student’s *t*-test (* *p* < 0.05; ** *p* < 0.01; ns, not significant).

## Data Availability

All the datasets supporting the conclusions of this article are included within the article and its Appendix A.

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
