# Peer review of "The MADS-Box Transcription Factor CaRIN Positively Regulates Chlorophyll Degradation During Pepper (*Capsicum annuum* L.) Fruit Ripening by Repressing the Expression of *CaLhcb-P4"

_plants, 2025, doi:10.3390/plants14030445_

Round 1
Reviewer 1 Report
Comments and Suggestions for Authors
The manuscript entitled ‘A MADS-box transcription factor CaRIN positively regulates chlorophyll degradation during pepper (Capsicum annuum L.) fruit ripening by repressing the expression of CaLhcb-P4’ addresses an important and novel aspect of pepper fruit ripening by elucidating the role of the MADS-box transcription factor CaRIN. The integration of transcriptomic and DNA affinity purification sequencing analyses, alongside functional assays, strengthens the study's findings. While the study is well-executed and the results are compelling, certain sections need improvement in clarity, particularly in methodology explanation and contextualization of findings.
The introduction provides a strong rationale for the study, emphasizing the economic and biological importance of fruit color in peppers. A clear knowledge gap in the regulatory mechanisms of pepper ripening was established. In the Materials and Methods section, the criteria for selecting the promoter region of CaLhcb-P4 need to be clarified and justified. Additionally, specify the statistical methods employed to ensure data reliability.
Specific comments:
Figure 1: the specificity of VIGS needs to be confirmed that no off-target effect that SlRIN homologs is silenced. In Figure 1D, the WL (#1) and UV (#2) looks like the same line, rather than two different lines. It is confusing whether pTRV2-GFP, pTRV2-CaPDS, or pTRV2-GFP-CaPDS is used as control.
Figure 2: which line (#1, #2 or both lines) was selected for transcriptome and subsequent analysis? Is there any discrepancy in line #1 and #2 in terms of gene expression profiles? Figure 2D and E, please indicate clearly the numbered samples are biological or technical replicates.
Figure 6: since CaRIN represses CaLhcb-P4 expression, please change the arrowhead to blunt end line for precise indication.
Author Response
#Response to Reviewer 1 Comments
Thank you very much for reviewing our manuscript. We greatly appreciate your thoughtful feedback. As you pointed out, there are several issues that need to be addressed. Based on your valuable suggestions, we have made extensive revisions to the previous draft. The detailed corrections are listed below.
Comments 1: In the Materials and Methods section, the criteria for selecting the promoter region of CaLhcb-P4 need to be clarified and justified.
Response 1: In the “4.6. Y1H Assay” of the MM section, we now state that the promoter fragments of CaLhcb-P4 (496bp) and CaUROS (447bp) containing the CaRIN binding peaks (up to 2-kb upstream from a TSS) were cloned into the pLacZi target vector to construct the pLacZi-CaLhcb-P4 and pLacZi-CaUROS recombinant vector. (Lines 391-392)
Similarly, in “4.7. Dual-luciferase Reporter Assay” of the MM section, we clarify that the same promoter fragment of CaLhcb-P4 (496bp) as in YIH was inserted into the pGreenII 0800-LUC vector (reporter vector). (Line 403)
Comments 2: Figure 1: the specificity of VIGS needs to be confirmed that no off-target effect that SlRIN homologs is silenced. In Figure 1D, the WL (#1) and UV (#2) looks like the same line, rather than two different lines. It is confusing whether pTRV2-GFP, pTRV2-CaPDS, or pTRV2-GFP-CaPDS is used as control.
Response 2: Regarding the specificity of VIGS, the gene Caz09g19480 in pepper has the highest sequence homology to SlRIN. Based on RNA-seq data, there was no significant difference in the transcript levels of this gene between the silenced fruits and the negative control, as shown in the figure below. This confirms that the VIGS approach was specific and did not cause off-target silencing of Caz09g19480.
In Figure 1D, "WL" and "UV" refer to photographs taken under different light conditions for the same plant. The labels (#1) and (#2) correspond to the fruits of two distinct lines in the second row, as indicated in the figure legend.
Regarding the control groups, pTRV2-GFP was used as the negative control, and pTRV2-CaPDS was used as the positive control. The label pTRV2-GFP-CaPDS in the figure was a typographical error, and we have corrected it in the revised version (Line 98).

Comments 3: Figure 2: which line (#1, #2 or both lines) was selected for transcriptome and subsequent analysis? Is there any discrepancy in line #1 and #2 in terms of gene expression profiles? Figure 2D and E, please indicate clearly the numbered samples are biological or technical replicates.
Response 3: Both lines #1 and #2 were selected for transcriptome and subsequent analyses. While there are slight differences in gene expression profiles between line #1 and line #2, the two lines generally cluster together and show a significant difference from the negative control.
In Figure 2D and E, we have clarified that the numbered samples represent biological replicates (Line 142).
Comments 4: Figure 6: since CaRIN represses CaLhcb-P4 expression, please change the arrowhead to blunt end line for precise indication.
Response 4: In response to the reviewer’s comment, we have replaced the arrowhead with a blunt-end line in Figure 6 to more accurately reflect the repression of CaLhcb-P4 expression by CaRIN.
Reviewer 2 Report
Comments and Suggestions for Authors
This manuscript investigates the genetic network governing color transformation during pepper fruit ripening, focusing on CaRIN through transcriptome analysis, yeast one-hybrid assays, and dual-luciferase reporter systems. The findings demonstrate that CaRIN positively regulates chlorophyll degradation by suppressing CaLhcb-P4 expression, providing novel insights into the regulatory mechanisms of chlorophyll degradation during pepper fruit maturation. The manuscript is well-written, with comprehensive research content, appropriate experimental design, standardized figures and tables, and accurate data analysis and statistics. In my assessment, it can be accepted for publication with minor revisions.
1. Lines 67-75 should be removed from the introduction section.
2. The manuscript needs to specify the dates when various databases were accessed for analysis.
3. Explain the calculation method of qRT-PCR.
4. Line 435, which data in the manuscript underwent significance testing at the 0.001 level?
Author Response
#Response to Reviewer 2 Comments
Thank you very much for reviewing our manuscript. We greatly appreciate your thoughtful feedback. As you pointed out, there are several issues that need to be addressed. Based on your valuable suggestions, we have made extensive revisions to the previous draft. The detailed corrections are listed below.
Comments 1: Lines 67-75 should be removed from the introduction section.
Response 1: We have removed Lines 67-75 from the manuscript. (Line 66)
Comments 2: The manuscript needs to specify the dates when various databases were accessed for analysis.
Response 2: In response to the reviewer’s comment, we have specified the dates when various databases were accessed for analysis. The transcription factors (TFs) with the highest sequence identity to CaRIN and other known MADS TFs were obtained from the following public databases: NCBI (https://www.ncbi.nlm.nih.gov, accessed on 12 November 2024) and the Pepper Genomics Database (http://ted.bti.cornell.edu/cgi-bin/pepper/index, accessed on 12 November 2024) for phylogenetic tree construction. Sequence alignment and phylogenetic tree construction were performed using ClustalX 2 (https://www.ebi.ac.uk/Tools/msa/clustalw2/, accessed on 12 November 2024) with 1000 bootstrap trials (Lines 339-343).
For selection of target sequences for VIGS, we used the Sol Genomics Network online tool (https://vigs.solgenomics.net/, accessed on 15 November 2024) (Line 348).
Additionally, we have included the version numbers of the software used in our analysis: MEME-chip software (v5.3.3) was employed to analyze the typical motifs in the peak regions [45]; peak-related genes were annotated using the R package ChIPseeker [46]; and the distribution of reads in the vicinity of the TSS was analyzed using depTools2 software (v3.3.0) (Lines 387-389).
Comments 3: Explain the calculation method of qRT-PCR.
Response 3: We have provided a more detailed explanation of the qRT-PCR calculation method. CaUBI-3 was used as the housekeeping gene, and the relative expression values were calculated using the 2^−ΔΔCT method (Lines 424-425).
Comments 4: Line 435, which data in the manuscript underwent significance testing at the 0.001 level?
Response 4: The transcript levels of CaRIN shown in Figure 1E underwent significance testing at the 0.001 level.
Round 2
Reviewer 1 Report
Comments and Suggestions for Authors
I have carefully reviewed the revised manuscript and the authors' responses to the comments. I am pleased to note that the authors have addressed all my concerns comprehensively, particularly the issues I raised regarding the figures。The authors have made significant efforts to revise and improve the quality of the figures, ensuring they are clear, accurate, and professionally presented.